# Aircraft-Type-Specific Impact of Speed Brakes on Lift and Drag

Judith Rosenow *, Thomas Sachwitz, Shumpei Kamo , Gong Chen and Hartmut Fricke

Institute of Logistics and Aviation, Technische Universität Dresden, 01069 Dresden, Germany;
thomas.sachwitz@tu-dresden.de (T.S.); shumpei.kamo@tu-dresden.de (S.K.); gong.chen1@tu-dresden.de (G.C.);
hartmut.fricke@tu-dresden.de (H.F.)
* Correspondence: Judith.Rosenow@tu-dresden.de

**Abstract:** The increasing influence of current research in air traffic management on daily flight operations leads to a stronger consideration of individually optimized aircraft trajectories. However, in the dichotomy between ecological, economic, and safety-based optimization goals, four-dimensionally optimized trajectories are subject to severe constraints in terms of position and speed. To fully assess the performance envelope of these trajectories, precise modelling of the influence of secondary control surfaces on flight performance is necessary. In particular, the use of speed brakes can significantly influence the descent and speed profile and allows the implementation of different cost indices. In this study, we present a modelling approach of the influence of extended speed brakes on flight performance and apply this method in a simulation environment for trajectory modelling of twelve different aircraft types. In doing so, we can determine an almost linear influence of the additional fuel requirement from the effective area of the speed brakes. The results can be implemented in any flight performance model and enable more precise modelling of future aircraft trajectories. Specifically, optimization targets regarding the required time of arrival, or the cost index and the consideration of the dynamic impact of atmospheric conditions in the trajectory optimization, only becomes possible through the calculation of the influence of the speed brake on lift and drag.

**Keywords:** speed brakes; aircraft trajectory optimization; flight performance calculation

## 1. Introduction

Due to the ever-increasing volume of air traffic and its concentration, especially in the airspaces around airports, new methods for increasing capacity are needed. Global crises, impending economic collapses, and the increasing social impact of aviation on global warming further require reducing the total costs and emissions of air traffic during flight. In addition to the technical optimization of existing aircraft models and the development of new modes of flying, changes to current flight operation procedures are also necessary to achieve these goals. An important step here is the change from static waypoint-fixed routes and approach procedures to individual continuous descent operations with routes adapted to the aircraft type, the atmospheric conditions, and the environment [1]. However, to calculate this most efficient flight path in advance, powerful flight performance models are necessary [2–4], which calculate these trajectories depending on weather data, airspace information, cost requirements and the performance data of the respective aircraft type. In real operations, however, the atmospheric conditions might deviate from those assumed in the flight performance calculations, and the pre-calculated Top of Descent (TOD) position might not completely follow the airline-specific optimization target anymore. An acceleration phase (for reducing or increasing speed) at an inefficient flight level might be necessary. For decelerating, speed brakes can be used with a significant impact on drag and a small impact on the lift. This effect, in turn, may, have a significant impact on fuel burn and is crucial for trajectory optimization. This impact will be shown in this paper.

The aim of this study is therefore to provide pilots with information on whether or how to return to the optimal flight path in the event of deviations from the flight path

calculated in advance, taking into account the performance characteristics of the speed brakes. Therefore, the aircraft-type-specific impact of speed brakes on flight behavior must be estimated by the flight performance model.

## 1.1. State of the Art

The impact of speed brakes on flight performance (i.e., on lift and drag) depends not only on aircraft-type-specific aerodynamic properties, but also on the speed and altitude of the aircraft. Furthermore, there are several different modes to adjust the speed brake position. Therewith, the degree of freedom increases significantly in flight performance calculations when considering speed brakes. Probably for this reason, the impact of speed brakes is often not in the focus [3,5–13] of the relevant models, or is approximated by a constant factor of $\Delta c_D = 0.02$ and $\Delta c_L = 0.00$ [14].

Older experimental studies focus on the effect of speed brake deflection on lift coefficient and pitching moment, because of their high relevance for controlling the aircraft. Although from the aerodynamic and flight performance point of view, the effect on the drag coefficient is more important, it has often not been measured and evaluated. Furthermore, experimental data only account for the pressure element of the drag when using absolute drag coefficients, ignoring the viscous contribution.

In his dissertation, Mclachlan [15] experimentally analyzed steady and unsteady flow fields generated by a typical two-dimensional transport airfoil-spoiler configuration without specification of the wing, and reported a decrease in lift coefficient $c_L$ up to $\Delta c_{L,\alpha=0} = -1.2$ for a speed brake deflection angle of $\delta = 60°$, and an angle of attack of $\alpha = 0°$. For an increasing angle of attack up to $\alpha = 16°$, the impact of speed brakes on lift coefficient decreases to $\Delta c_{L,\alpha=16} = -0.8$.

Stucky [16] demonstrates the use of a modified step-wise regression technique for estimating the aerodynamic nonlinearities inherent in lateral spoiler control and claims that, as compared to conventional control surfaces, wing spoilers exhibit higher-order characteristics when used in the roll. These features were represented using polynomial splines as functions of angle-of-attack and spoiler deflection. Unfortunately, the author only considered the lateral response of the aircraft model and therewith concentrated on the impact of spoiler deflection angle and angle of attack on rolling $c_{Mx}$ and yawing moment coefficients. He proved assumed linear relationships between the resultant side-forces and the spoiler deflection angle of attack with real-flight measurements.

Abdelrahman et al. [17] experimentally studied the effects of spoiler configurations on the aircraft aerodynamics of a B-747 airplane under wind shear conditions and show that spoiler deflection enhances the wind shear-induced lift loss, especially at low angles of attack. They published experimentally gained values of $\Delta c_{L,\alpha=16,\delta=90} = 1.25$, corresponding to $-70\%$ of $c_{L,\alpha=16,\delta=0}$ and $\Delta c_{L,\alpha=4,\delta=90} = 0.6$ which is $-1\%$ of $c_{L,\alpha=4,\delta=0}$. Wind shear tends to amplify the spoiler effect on the side force and yawing moment, but only at high angles of attack does it alter the resulting rolling moment. Furthermore, Abdelrahman proved that speed brake deflections, on the other hand, have almost no influence on the aircraft stability margin, which is generally increased when wind shear is present [17].

Lindsay et al. [18] experimentally proved that at high angles of attack ($\alpha > 16°$) and high deflection angles ($\delta > 25°$), the spoiler airfoil on top of a NACA 2412 airfoil created even less drag and more lift than in a clean configuration. Beyond these extreme conditions, they gained similar results for the increase in a lift ($\Delta c_L = -0.375$ and $\Delta c_D = 0.05$). However, the results do not depend on altitude and speed and are not formalized to be used for other than the plotted values.

Geisbauer [19] published results of fluid dynamic simulations of steady and unsteady simulations of static and dynamic spoiler deflections in the low-speed regime with an in-house flow solver TAU, modelling an in-house DLR-F15DS (dynamic spoiler) model which is an assembly of a 2D wall-to-wall model representing a small aircraft wing. He also investigated the dynamic impact of speed brake deflection on lift and drag and gained results in the range of $\Delta c_{L,\alpha=0,\delta=60} = -0.35$ and $\Delta c_{D,\alpha=0,\delta=60} = 0.05$.

All references mentioned so far focus on the discussion of experimental studies without transferring the results into equations and thus, applying them to data other than the tested input data. However, all experiments show a greater influence of speed brakes on lift $\Delta c_L = 0.00$ and drag $\Delta c_D = 0.02$ than assumed by [14]. From this it follows that the results can be used for validation purposes and comparisons with the theoretically calculated values.

Omori [20] also presented experimental results of a spoiler as a panel vertically standing on a flat wing surface, and calculated the lift coefficient decrease due to speed brake deflection. He used the speed brake height and the speed brake location as dependent variables and parameterized a reference deflection angle $\theta_0$ and an effective height $h$. The theoretical results were validated by measurements on a NACA 0009 airfoil. However, due to unknown profile geometry, speed and altitudes, the results are not applicable to other aircraft types and flight conditions.

In 1966, Barnes [21] developed an experimental approach for measuring the impact of speed brakes on lift and pitching moment on RAE100 and RAE102 airfoils. To do so, he identified dependencies of $c_L$ on the boundary layer displacement thickness on the aerofoil at the position of the speed brake. Therewith, he calculated $\Delta c_L = -0.2$. The RAE airfoils, however, seem to have lower maximum values of $c_{L,max} = 1.0$. From this, it follows that the method might not be applicable to large modern aircraft types. Additionally, the method described in this paper depends on several parameters provided by hand-printed graphs in low resolution with a high degree of uncertainty.

Kalligas [22] and Lee [23] looked at static, while Consigny et al. [24], Costes et al. [25], and Nelson et al. [26] considered harmonic oscillation in their experiments, and Yeung et al. [27] investigated ramp-type spoiler deflections in two dimensions. Consigny et al. [24] discovered that, unlike deflection, quick retraction of a spoiler had no negative consequences. Kalligas [22] observed that the drag response is slower than the lift response.

Finally, three-dimensional spoilers have been experimentally studied by Jordan et al. [28] and Scott et al. [29]. However, the experiments were applied to very old mid-wing electronic-warfare aircraft models [28] with aerodynamics different from today's civil aircraft types. Scott et al. [29] only published the drag polar for a NACA airfoil 0012, without showing the effect of spoiler deflection on a lift and drag individually.

From the literature studied so far, it follows that either unsuitable aircraft types were considered or important dependencies (such as speed and altitude) are missing in the experimental setups and published values of $\Delta c_L$ and $\Delta c_D$. The results do not make it possible to elaborate a general model for the impact of speed brake deflection on lift and drag.

In 1971, Hanke [30] quantified the impact of spoilers on flight performance (i.e., the impact on lift and drag) in a granular manner, for a research project of the National Aeronautics and Space Administration (NASA) on a modern aircraft type (B747-100). In this study, this NASA model is transferred to twelve aircraft types.

*1.2. Paper Structure*

After the introduction, Section 2 provides an overview of forces considered in flight performance and some theoretical considerations of speed brakes in this system of equations. Section 3 describes the reference speed brakes model for the B747 and important geometrical and aerodynamic details of the B747, before the transfer of the reference speed brake model to other aircraft types is described and discussed. The implementation of the adapted speed brake model is shown in Section 4. Using a deceleration phase just above 10,000 ft, the influence of speed brakes on the four-dimensional descent profile and the resulting fuel consumption is analyzed for different aircraft types. In Section 5, the paper concludes with a summary and conclusion.

## 2. Flight Performance Calculation

Aircraft movements can be described by forces acting on the aircraft. In the two-dimensional view, these essentially comprise lift $F_L$ (Equation (1)), weight $F_G$ (Equation (2)), drag $F_D$ (Equation (3)), and thrust $F_T$ (Equation (4)) forces, as shown in Figure 1.

$$F_L = \frac{\rho}{2} v_{\text{TAS}}^2 \, S \, c_L \tag{1}$$

$$F_G = m \, g \tag{2}$$

$$F_D = \frac{\rho}{2} v_{\text{TAS}}^2 \, S \, c_D \tag{3}$$

$$F_T \cos \gamma - F_L \sin \gamma - F_D \cos \gamma = m \, a_x \tag{4}$$

where $v_{\text{TAS}}$ is the true air speed [m s$^{-1}$], $m$ the aircraft mass [kg], and $a_x$ the acceleration [m s$^{-2}$] in the horizontal plane $_x$. $c_D$ and $c_L$ denote drag coefficient [a.u.] and lift coefficient [a.u.], and $\rho$ describes air density [kg m$^{-3}$]. $S$ [m$^2$] describes the total wing area (including non-deflected control surfaces) and $\gamma$ [rad] the flight path angle [31]. During the flight, the forces act on different positions of the aircraft, resulting in additional moments to the vertical and lateral axes. Only in an equilibrium state of all forces and moments, does the aircraft fly straight with constant speed. Control surfaces are used to change the magnitude and the points of application of those forces, producing moments along the three axes and controlling aircraft rotation.

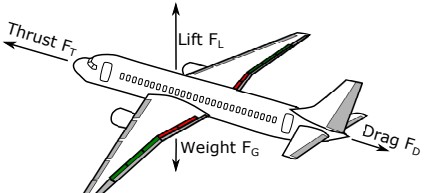

**Figure 1.** Forces acting on an aircraft in the flight-path-oriented coordinate system. Thrust $F_T$ and drag $F_D$ form the *x*-axis.

Depending on the flight attitude, lift and also drag may need to be changed. Therefore, secondary control surfaces are installed on commercial aircraft (see Figure 2), which essentially comprise the landing flaps and speed brakes [32]. Speed brakes or spoiler flaps are movable flaps on the upper wing surface and are not primarily used for aircraft rotation around the three axes (see Figure 2 for details). In modern commercial aircraft, spoilers are also used to support the primary control surfaces, which is why they are now differently referred to as ground spoilers, speed brakes or roll spoilers [33], depending on their function and use [32]. Extended secondary control surfaces increase the drag on the wing and thus make the aircraft descend if the magnitude of thrust is maintained. A single panel can also perform several functionalities, depending on the aircraft's current attitude and flight phase. In the following, different uses of secondary control surfaces during the flight are described.

Some of the secondary control surfaces can be manually extended as speed brakes, ensuring a symmetrical deflection of the panels. Speed brakes can be extended either individually or altogether. According to the deflection, speed brakes increase the drag (which results in a corresponding reduction in speed) and decrease lift [33]. The effect of an individually extended speed brake *i* on drag and lift can be described as a change in drag coefficient $\Delta c_{D,i}$ [a.u.] and lift coefficient $\Delta c_{L,i}$ [a.u.]. By using the speed brakes, steeper flight profiles can be flown without exceeding the speed limits of the aircraft type, or the airspace currently being flown through [32]. The angle of the speed brakes can be set on all commercial aircraft using a lever in the cockpit. The principle for speed-brake setting differs not only according to aircraft type and position on the wing, but also depending on the manufacturer as concerns the type of activation. For example, the manufacturers

Airbus and Bombardier have fixed detents for the various angles on the selection lever for the speed brakes [34,35], while Boeing and Embraer allow the pilot to adjust them continuously [36,37].

Most of the secondary control surfaces acting as roll spoilers are used to support the ailerons to increase roll moment along the longitudinal axis to initiate a turn manoeuvre. Since the spoiler panels serve to reduce lift, they are extended on one side of the wing for a turn manoeuvre. The resulting asymmetric lift causes the wing to sink on one side, which creates a moment about the longitudinal axis. As roll spoilers have two functionalities, one as spoilers and the other as ailerons, they are also called spoilerons. Unlike ailerons, however, spoilers can only be extended upwards, and thus only reduce lift and do not increase it [32]. Since modern flight control systems always combine yaw and roll momentum to initiate a turn, the spoilers also take over part of the function of the rudder, so that it does not have to be additionally deflected to initiate the turn [32].

Ground spoilers are extended to their maximum angle during landing to reduce lift as much as possible. The additional drag caused by the ground spoilers, on the other hand, plays only a small role during landing [32]. The lift on the wing in the area of the spoilers is eliminated since the airflow can no longer be applied here. During so-called firm landing procedures, the landing gear is pressed more strongly onto the ground for increasing the efficiency of the brakes [32]. After touch-down, it is important to reduce as much lift as possible as quickly as possible so that all existing spoiler panels are usually extended to their maximum angle as ground spoilers.

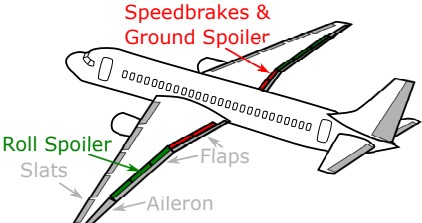

**Figure 2.** Primary (grey) and secondary (red, green) flight control surfaces on a commercial aircraft.

Modern transport aircraft use flap spoilers: a flap spoiler is a panel affixed to the wing upper surface trailing edge region that, when deflected upwards (the panel rotating about its leading edge), causes the flow to separate over the wing surface in a controlled manner, resulting in a decrease in lift and an increase in drag. Due to the current inability to simulate separated flows, speed brake aerodynamic properties are the most challenging of the airplane control surfaces to anticipate. Speed brakes have a number of characteristics that make them ideal for lateral control in aircraft: large rolling moments are produced by speed brakes, and speed brakes are an alternative to ailerons for full roll control, allowing full span flaps to be used, with apparent STOL uses and benefits. Ailerons provide an undesirable yawing moment, whereas speed brakes produce a favourable yawing moment. Furthermore, speed brakes, as opposed to ailerons, are usually more effective at high speeds and are less prone to suffer from aeroelastic effects [15].

Unfortunately, several of the aerodynamic characteristics of spoilers increase the complexity of predicting the unsteady flow field generated by airfoils with deflected speed brakes. The speed brake control effectiveness is non-linear: the lift decrease is a non-linear function of the speed brake deflection. This non-linearity is most noticeable when speed brakes are employed in conjunction with a deflected flap. This requires the speed brakes to be integrated with other control surfaces (such as ailerons) in order to give linear control (necessary to satisfy the pilot and autopilot functions). Historically, this feature has made it difficult to achieve the use of speed brakes for full roll control in the presence of full span flaps. When the speed brake is deflected, the resulting turbulent wake is extremely unsteady. The wing interacts with the horizontal tailor and buffets (i.e., aerodynamics-induced vibrations) can be caused by themselves. Furthermore, the time delay between the deflection of the spoiler and the reduction in lift causes a delay in the aircraft's response to

speed brake deflection. Finally, the change in wing pitching-moment with spoiler deflection, as well as the influence of the spoiler wake on the horizontal tail, can generate unacceptable pitching moments.

*Preliminary Considerations of Speed Brakes*

The extension of the individual panels causes an increase in drag, as the surfaces are pressed into the air flowing around them. The impact of speed brakes on forces can be reduced to that on the coefficients of the forces. Mainly, speed brakes influence flight performance through four different coefficients: the lift coefficient $c_L$, drag coefficient $c_D$, pitch moment coefficient $c_{My}$, and the roll moment coefficient $c_{Mx}$.

The total change in drag $\Delta F_D$ [N] is the sum of the impact of each control surface, where $n$ denotes the number of speed-brake panels

$$\Delta F_D = \sum_{i=1}^{n} \Delta F_{D,i}. \tag{5}$$

Assuming constant speed $v$ [m s$^{-1}$], wing area $S$ [m$^2$], and air density $\rho$ [kg m$^{-3}$], the change in drag also results in a change in the drag coefficient $c_D$ [38]. Since all parameters except for the drag force are constant on the entire aircraft, the change in the total drag coefficient can also be calculated by

$$\Delta c_D = \frac{2\,\Delta F_D}{v^2\,\rho\,A} = \frac{2\,\sum_{i=1}^{n} \Delta F_{D,i}}{v^2\,\rho\,A}. \tag{6}$$

The extension of speed brakes also causes a reduction of the lift at the position of the extended speed brakes, since in these areas of the wing the flow no longer completely flows around the profile. The lift is even omitted with a correspondingly large extension angle. The sum of the individual changes in lift forces $\Delta F_{L,i}$ [N] results in the total change in lift $\Delta F_L$ [N]

$$\Delta F_L = \sum_{i=1}^{n} \Delta F_{L,i} \tag{7}$$

As with drag, changes in the lift coefficient $c_L$ can be calculated from the lift change. Since the speed, air density, and wing area can be assumed to be uniform for the entire aircraft, the change in the total lift coefficient can also be calculated in this way [38]

$$\Delta c_L = \frac{2\,\Delta F_L}{v^2\,\rho\,A} = \frac{2\,\sum_{i=1}^{n} \Delta F_{L,i}}{v^2\,\rho\,A}. \tag{8}$$

The moment around the pitch axis is also influenced by the use of the speed brakes. An increase in drag and a reduction in lift at the location of the speed brake (with longitudinal distance $d_{y,i}$ [m] to the point of action of the force on the $i$-th control surface) generates a moment about the average point of application of lift on the aircraft. Thus, the change in pitch moment $\Delta M_y$ can be calculated by [32]

$$\Delta M_y = \sum_{i=1}^{n} \left( \Delta F_{L,i}\, d_{y,i} \right) \tag{9}$$

The change in the pitch moment coefficient $\Delta c_{My}$ depends on the reference wing depth $l_\mu$ [m]

$$\Delta c_{My} = \frac{2\,\Delta M_y}{v^2\,\rho\,A\,l_\mu} = \frac{2\,\sum_{i=1}^{n} \left( F_{L,i}\, d_{y,i} \right)}{v^2\,\rho\,A\,l_\mu}. \tag{10}$$

The roll moment $M_x$ only affects the spoiler panels, which are used as roll spoilers. By reducing the lift at the location of the spoiler, the wing sinks on that side, which provides a moment along the longitudinal axis of the aircraft. The change in roll moment $\Delta M_x$ can

be calculated according to the same principle as the change in pitch moment by multiplying the change in the lift force $\Delta F_{L,i}$ by the respective distance $d_{x,i}$ to the longitudinal axis

$$\Delta M_x = \sum_{i=1}^{n} (\Delta F_{L,i}\, d_{x,i}). \tag{11}$$

The change in the roll moment coefficient $\Delta c_{Mx}$ again depends on the reference wing depth $l_\mu$ [m]

$$\Delta c_{Mx} = \frac{2\,\Delta M_x}{v^2\,\rho\,A\,l_\mu} = \frac{2\,\sum_{i=1}^{n}(F_{L,i}\,d_{x,i})}{v^2\,\rho\,A\,l_\mu}. \tag{12}$$

## 3. Methodology

### 3.1. Speed Brakes on NASA Model B747

The B747 consists of six speed brakes per wing (see Figure 3). Six speed brakes per wing are extended synchronously, whereby the inner sped brakes can only be extended by 20° compared to the outer speed brakes with a maximum angle of 45° (see Table 1). In addition to the theoretical calculation already explained, Hanke[30] calculates the modified coefficients for each speed brake panel $i$ separately using Boeing-specific performance coefficients. These coefficients depend on speed brake angle $\delta_i$, altitude, angle of attack $\alpha$, flap configuration, and speed. The values are digitized based on the graphic representation by Hanke. We interpolate these coefficients between provided discretized values with increments of 10,000 feet for altitude, 5° for speed brake angle, and 5° for the angle of attack, as well as for Mach numbers (0.5 < Mach < 0.85) with $\Delta$ Mach = 0.1.

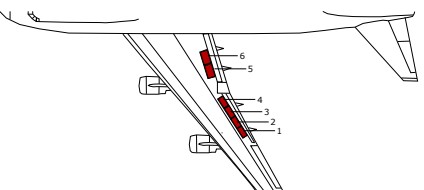

**Figure 3.** Assumed number and position of spoilers on a single B747 wing.

The change in the lift coefficient for the $i$-th panel $\Delta c_{L,i}$ is expressed as

$$\Delta c_{L,i} = K_{\delta,i}(\Delta c_{L,i})_{45}\, \frac{(c_{L,i})_M}{(c_{L,i})_{M_0}} \left( \frac{L_{E,i}}{L_{R,i}} \right) F_{L,GE} \tag{13}$$

where $K_{\delta,i}$ determines speed-brake-specific effectiveness factors and $(\Delta c_{L,i})_{45}$ describes the change in lift coefficient due to a speed brake extension to 45°. Speed effects are considered with $\frac{(c_{L,i})_M}{(c_{L,i})_{M_0}}$ as change in the lift coefficient due to Mach number $M$ compared to the change in the lift coefficient at $M = 0$ (symbol $M_0$ in Equation (13)). Since there is no change in the lift coefficient at $M = 0$, the ratio takes the value 1 at $M = 0$. The aeroelastic effect is considered with $\left( \frac{L_{E,i}}{L_{R,i}} \right)$. The ground effect factor $K_{GE}^B$ only effects $\Delta c_{L,i}$ below an altitude of 90 feet and therefore we assume $K_{GE}^B = 0$. Below 90 feet we assume $F_{L,GE} = 1$ [30].

With $K_{GE}^B = 0$, the change in drag coefficient $\Delta c_{D,i}$ depends on the change of the drag coefficient by extending the speed brakes at an angle of attack of 4° ($\Delta c_{D,i}(\alpha = 4)$) and on the rate of change of drag coefficient with angle of attack $\frac{d(\Delta c_{D,i})}{d\alpha}$ [30].

$$\Delta c_{D,i} = \left[ \Delta c_{D,i}(\alpha = 4) + \frac{d(\Delta c_{D,i})}{d\alpha}(\alpha - 4) \right] \frac{(c_{D,i})_M}{(c_{D,i})_{M_0}} \tag{14}$$

Assuming $K_{GE}^B = 0$ and $F_{m,GE} = 1$, the change of the pitch moment coefficient $\Delta c_{My,i}$ due to extended speed brake $i$ is defined as [30]

$$\Delta c_{My,i} = (K_{\delta,i})_m (\Delta c_{m,i})_{45} \frac{(c_{m,i})_M}{(c_{m,i})_{M_0}} \left( \frac{M_{E,i}}{M_{R,i}} \right) \tag{15}$$

where $(K_{\delta,i})_m$ describes the effectiveness of the respective speed brake panel and $(\Delta c_{m,i})_{45}$ the change in the basic pitching moment, with extended speed brakes by 45°. Again, the ratio $\frac{(c_{m,i})_M}{(c_{m,i})_{M_0}}$ refers to the effect of the change in Mach number. The aeroelastic effect is considered by the ratio $\left( \frac{M_{E,i}}{M_{R,i}} \right)$ for each speed brake $i$.

The same procedure is used for the change in the roll moment coefficient $\Delta c_{Mx,i}$ with extended speed brake $i$ [30]:

$$\Delta c_{Mx,i} = (K_{\delta,i})_l (\Delta c_{l,i})_{45} \frac{(c_{l,i})_M}{(c_{l,i})_{M_0}} \left( \frac{R_{E,i}}{R_{R,i}} \right) \tag{16}$$

$\Delta c_{Mx,i}$ depends on the effectiveness of the individual speed brake panel $(K_{\delta,i})_l$ and on the change in the roll moment coefficient $(\Delta c_{l,i})_{45}$ by extending speed brake $i$ panel to 45°. The effect of Mach number is described by the component $\frac{(c_{l,i})_M}{(c_{l,i})_{M_0}}$ and the aeroelastic effect on the rolling moment coefficient is described by the component $\left( \frac{R_{E,i}}{R_{R,i}} \right)$.

After the impact of an extended speed brake $i$ on flight performance has been obtained for the B747, we determine the corresponding effective panel area of the B747 for different aircraft types (using Equation (18)) and transfer the model, assuming a similar impact on lift, drag, and moment for a similar effective panel area. The aerodynamic properties of the wing and thus the properties of the spoiler are determined by the airfoil shape, i.e., the depth and thickness of the wing. Since the maximum lift of an airfoil shape is mainly determined by the maximum thickness and camber of an airfoil [39], the maximum airfoil thickness of the wing is chosen as the reference value for comparing speed brake panels as an averaged value in the middle of the speed brake width $l_{w,i}$ [m]. This assumption is underlined by the test series of NASA on the flow influence of speed brakes [40]. Here it was found that an extended speed brake does not influence the pressure ratio on the airfoil surface in front of the speed brake [40]. Thus, it can be assumed that the forward airfoil continues to maintain its effectiveness in terms of maximum lift. Here, as in other tests [41], there is only a reduction in the maximum possible lift. Further tests by NASA in the subsonic range up to a Mach number of 0.94 result in a speed brake-induced loss of lift between $\Delta c_L = 0.0015$ and $\Delta c_L = 0.0025$ for small and large extension angles, respectively [42]. Considering the definition of the Reynolds number $Re$

$$Re = \frac{\rho\, v\, l}{\eta} \tag{17}$$

(where $\rho$, $v$, $l$ and $\eta$ are the fluid density [kg m$^{-3}$], the fluid speed [m s$^{-1}$], the characteristic linear dimension [m], and the dynamic viscosity of the fluid [N s m$^{-2}$], respectively) it can be concluded that when comparing different speed brake panels, the wing depth at this point has little effect on the speed brake impact and the variation of the speed brake angle is more relevant [42]. Note that this fact is only valid for laminar flows adjacent to the profile in the subsonic range, a constant angle of attack $\alpha$, and a clean flap configuration. Furthermore, the maximum speed must be well below the speed of sound (below Mach 0.94) [30].

To comparatively determine the speed brake of a different aircraft type, the geometric dimensions of the speed brake panel, as well as the profile thickness, are determined.

Subsequently, the speed brake angle $\delta_{i,\text{B747}}$ is calculated for the respective B747 reference panel with which it reaches the same effective area $A_i$ [m$^2$]

$$A_i = b_i l_i \sin \delta_{i,\text{B747}} \tag{18}$$

where $b_i$ and $l_i$ are the speed brake width [m] and speed brake length [m], respectively. Using these angles, the variables for the coefficients in Equations (6), (13), (15) and (16) are calculated.

### 3.2. Geometric Data of NASA Model B747

For the B747, the following geometric data with impact on speed-brake-specific flight performance is assumed.

**Table 1.** Assumed geometric dimensions of the B747-100 speed brakes (Data collected from [43]). The Panels are numbered from outside to inside (see Figure 3).

| Panel Number | Length $l_i$ [m] | Width $b_i$ [m] | Max. Angle $\delta_i$ [°] | Max. Area $A_i$ [m$^2$] | Mean Wing Depth $l_{w,i}$ [m] |
|---|---|---|---|---|---|
| 1 | 1.109 | 1.905 | 45 | 1.495 | 0.37 |
| 2 | 1.109 | 1.905 | 45 | 1.495 | 0.377 |
| 3 | 1.109 | 1.905 | 45 | 1.495 | 0.488 |
| 4 | 1.109 | 1.905 | 45 | 1.495 | 0.559 |
| 5 | 1.397 | 2.286 | 20 | 1.092 | 0.936 |
| 6 | 1.397 | 2.286 | 20 | 1.092 | 1.187 |

### 3.3. Transfer of the B747 Model to Other Aircraft Types

The following aircraft types (and data sources) are considered in this study: the Airbus models A310 [44], A319 [44,45], A320 [44,45], A321 [44,45], A330 [44,46] and A380 [44,47]; the Boeing models B737 [48], B767 [49,50], B777 [36,51–53]; Bombardier CRJ900 [54,55]; and Embraer E170 [37] and E190 [56]. Geometric information for panel sizes is taken from [44], whereas operational information (i.e., possible configurations for speed brake and maximum speed brake angle) are taken from aircraft operational manuals. Because the E190 and the E170 have identical performance data, wing and speed brake dimensions (see [56]), there should be no differences in speed brakes on lift and drag. During the literature research, a few special features stood out. In comparison to Boeing aircraft types, where speed brake angles $\delta_i$ can take arbitrary values between $\delta_i = 0°$ and $\delta_{i,max}$, Airbus, Bombardier and Embraer only enable discrete values of $\delta_i$ in full, three quarter, half and quarter positions [34], except for A310, which has 11 positions [35]. Note that only those panels used as speed brakes are listed in Tables 2 and 3.

In Tables 2 and 3, aircraft-specific speed brake information is summarized. Due to the geometry, we limited the list to a single wing side. The speed brakes are recorded according to the manufacturer-specific numbering. Airbus numbers the speed brake from inside to outside, whereas Boeing, Bombardier, and Embraer number the speed brake from outside to inside. Besides the number of speed brakes per wing, the mean wing depth $l_{w,1}$ at the position of each speed brake allows the allocation of each speed brake to a corresponding B747 speed brake. The individual aircraft speed brake length $l_i$ and width $b_i$ are used to calculate the individual speed brake area $A_i$. The individual speed brake area in turn is used to estimate the corresponding B747 speed brake angle $\delta_{i,\text{B747}}$. The described speed brake panel in Tables 2 and 3 are limited to those spoilers with speed brake function. For example, the inner and outer panels No. 1 and No. 5 of A319 are not used as speed brakes [45] and panel 4 of B767 is only used at low speeds and not relevant for this paper [50].

**Table 2.** Transfer of Airbus-specific speed brakes and their mean wing depth $l_w, i$ to the speed brakes of the B747 (No, B747) parameterized by NASA (numbered from outside to inside). Additionally, Airbus speed brake length $l_i$, width $b_i$, area $A_i$, maximum speed brake angles $\delta_{i,max}$, and the resulting maximum speed brake angles of the corresponding B747 speed brake $\delta_i$ [°] are given.

| Panel | $b_i$ [m] | $l_i$ [m] | $\delta_i$ max [°] | $A_i$ max [°] | No B747 | $l_{w,i}$ [m] | $\delta_i$ B747 [°] |
|---|---|---|---|---|---|---|---|
| **A310** | | | | | | | |
| 1 | 1.62 | 0.743 | 35 | 0.69 | 5 | 1.006 | 12.485 |
| 2 | 1.69 | 0.904 | 35 | 0.876 | 4 | 0.746 | 24.488 |
| 3 | 1.87 | 0.684 | 35 | 0.734 | 2 | 0.428 | 20.31 |
| 4 | 1.862 | 0.684 | 35 | 0.731 | 2 | 0.41 | 20.277 |
| 5 | 1.526 | 0.549 | 0 | 0.286 | 2 | 0.388 | 7.786 |
| 6 | 1.529 | 0.549 | 0 | 0.287 | 1 | 0.377 | 7.805 |
| 7 | 1.478 | 0.549 | 0 | 0.277 | 1 | 0.289 | 7.542 |
| **A319** | | | | | | | |
| 1 | 1.763 | 0.626 | 17.5 | 0.633 | 2 | 0.381 | 17.434 |
| 2 | 1.661 | 0.625 | 25 | 0.596 | 1 | 0.297 | 16.37 |
| 3 | 1.528 | 0.625 | 25 | 0.548 | 1 | 0.343 | 15.016 |
| 4 | 1.531 | 0.627 | 0 | 0.55 | 1 | 0.251 | 15.096 |
| **A320** | | | | | | | |
| 1 | 1.763 | 0.626 | 12.5 | 0.633 | 2 | 0.381 | 17.434 |
| 2 | 1.661 | 0.625 | 25 | 0.596 | 1 | 0.343 | 16.37 |
| 3 | 1.528 | 0.625 | 25 | 0.548 | 1 | 0.297 | 15.016 |
| 4 | 1.531 | 0.627 | 0 | 0.55 | 1 | 0.251 | 15.096 |
| **A321** | | | | | | | |
| 1 | 1.763 | 0.626 | 25 | 0.633 | 2 | 0.381 | 17.434 |
| 2 | 1.661 | 0.625 | 25 | 0.438 | 1 | 0.343 | 11.986 |
| 3 | 1.528 | 0.625 | 25 | 0.548 | 1 | 0.297 | 15.016 |
| 4 | 1.531 | 0.627 | 0 | 0.55 | 1 | 0.251 | 15.096 |
| **A330** | | | | | | | |
| 1 | 2.43 | 0.693 | 25 | 0.711 | 5 | 1.054 | 12.871 |
| 2 | 2.12 | 0.705 | 30 | 0.857 | 4 | 0.682 | 23.913 |
| 3 | 2.163 | 0.707 | 30 | 0.878 | 4 | 0.597 | 24.531 |
| 4 | 2.165 | 0.709 | 30 | 0.88 | 3 | 0.541 | 24.614 |
| 5 | 2.193 | 0.71 | 30 | 0.893 | 3 | 0.49 | 24.998 |
| 6 | 2.209 | 0.712 | 30 | 0.902 | 2 | 0.445 | 25.254 |
| **A380** | | | | | | | |
| 1 | 2.43 | 1.296 | 20 | 1.077 | 6 | 1.655 | 19.713 |
| 2 | 2.374 | 1.229 | 20 | 0.998 | 6 | 1.34 | 18.211 |
| 3 | 1.972 | 1.117 | 20 | 0.754 | 5 | 1.154 | 13.649 |
| 4 | 2.195 | 1.095 | 20 | 0.882 | 5 | 1.042 | 14.919 |
| 5 | 2.084 | 1.056 | 20 | 0.752 | 5 | 0.985 | 13.629 |
| 6 | 2.782 | 0.883 | 45 | 1.736 | 4 | 0.907 | 55.241 |
| 7 | 2.503 | 0.737 | 45 | 1.305 | 4 | 0.878 | 38.13 |
| 8 | 2.35 | 0.581 | 45 | 1.329 | 4 | 0.788 | 38.956 |

The determination of the change in the roll moment coefficient $\Delta c_{Mx,i}$ is made by taking into account the corresponding maximum B747 deflection angle $\delta_{i,max}$ to support the rolling function. This may differ from the maximum available angle when used as a speed brake.

**Table 3.** Transfer of aircraft-specific speed brakes and their mean wing depth $l_{w,i}$ to the speed brakes of B747 (No, B747) parameterized by NASA (numbered from outside to inside). Additionally, individual aircraft speed brake length $l_i$, width $b_i$, area $A_i$, maximum speed brake angles $\delta_{i,max}$, and the resulting maximum speed brake angles of the corresponding B747 speed brake $\delta_i$ [°] are given. Note that the E190 and the E170 have the same performance data as well as wing and speed brake dimensions [56], so no differences in speed brakes on lift and drag can be expected.

| Panel | $b_i$ [m] | $l_i$ [m] | $\delta_i$ max [°] | $A_i$ max [°] | No B747 | $l_{w,i}$ [m] | $\delta_i$ B747 [°] |
|---|---|---|---|---|---|---|---|
| **B737** | | | | | | | |
| 2 | 1.05 | 0.567 | 19.5 | 0.199 | 3 | 0.499 | 4.495 |
| 3 | 1.5 | 0.594 | 19.5 | 0.297 | 3 | 0.509 | 6.732 |
| 4 | 1.012 | 0.612 | 24.5 | 0.257 | 3 | 0.538 | 5.807 |
| 5 | 1.008 | 0.632 | 24.5 | 0.264 | 3 | 0.529 | 5.973 |
| **B767** | | | | | | | |
| 1 | 1.532 | 0.736 | 45 | 0.797 | 3 | 0.489 | 18.318 |
| 2 | 1.532 | 0.736 | 45 | 0.797 | 3 | 0.537 | 18.318 |
| 3 | 1.532 | 0.736 | 45 | 0.797 | 4 | 0.58 | 18.318 |
| 5 | 1.768 | 0.952 | 17 | 0.492 | 4 | 0.93 | 11.192 |
| 6 | 1.684 | 0.952 | 17 | 0.496 | 5 | 1.059 | 8.04 |
| **B777** | | | | | | | |
| 1 | 2.032 | 0.642 | 60 | 1.13 | 4 | 0.559 | 32.329 |
| 2 | 1.95 | 0.641 | 60 | 1.082 | 4 | 0.62 | 30.795 |
| 3 | 1.992 | 0.655 | 60 | 1.13 | 4 | 0.681 | 32.305 |
| 4 | 1.91 | 0.661 | 45 | 0.893 | 4 | 0.729 | 24.991 |
| 5 | 1.958 | 0.667 | 0 | 1.131 | 4 | 0.802 | 32.347 |
| 6 | 1.995 | 1.011 | 60 | 1.747 | 6 | 1.252 | 33.172 |
| 7 | 1.995 | 1.011 | 60 | 1.747 | 6 | 1.562 | 33.172 |
| **CRJ900** | | | | | | | |
| 1 | 0.815 | 0.199 | 50 | 0.124 | 1 | 0.3 | 3.373 |
| 2 | 0.85 | 0.199 | 50 | 0.13 | 1 | 0.268 | 3.519 |
| **E170/E190** | | | | | | | |
| 3 | 1.433 | 0.712 | 30 | 0.521 | 1 | 0.34 | 9.864 |
| 4 | 1.309 | 0.655 | 30 | 0.439 | 1 | 0.317 | 8.291 |
| 5 | 1.122 | 0.616 | 30 | 0.353 | 1 | 0.274 | 6.658 |

Figure 4 depicts the change in the lift coefficient $\Delta c_{L,i}$ and the drag coefficient $\Delta c_{D,i}$ for an Airbus A310 aircraft. depending on speed brake angle, at 30,000 feet altitude and Mach 0.82, with the flaps in clean configuration and the angle of attack at $\alpha = 0°$. As expected, the total lift coefficient decreases and the drag coefficient slightly increases with increasing speed brake angle. Since Airbus numbers speed brakes from inside to outside, the effect of inner speed brakes is larger than the effect of outer speed brakes. Furthermore, the effect of extended speed brake on $\Delta c_{L,i}$ and $\Delta c_{D,i}$ increases with increasing speed brake angle $\delta_i$.

For the twelve aircraft types mentioned, it was possible to determine the change in drag coefficients and lift coefficients, which can be classified in the respective performance field of the aircraft types and thus allow correspondingly good comparison possibilities of the individual types (see Figure 5 for maximum values).

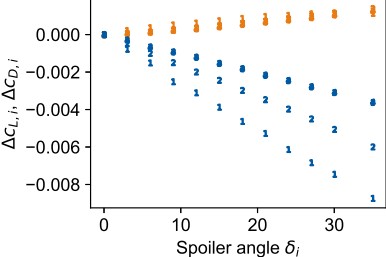

**Figure 4.** Change in lift coefficient $\Delta c_{L,i}$ (blue) and drag coefficient $\Delta c_{D,i}$ (orange) for different speed brake angles of an Airbus A310 aircraft. A clean flap configuration, a Mach number of 0.82, an angle of attack of $\alpha = 0°$, and an altitude of 30,000 ft are assumed. A310 speed brakes are numbered from inside to outside (scatter markers).

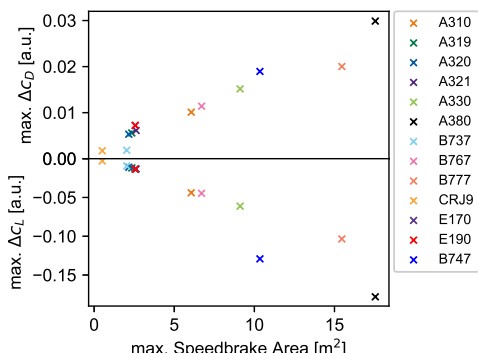

**Figure 5.** Change in maximum total drag coefficient $\Delta c_D$ (**top**) and maximum total lift coefficient $\Delta c_L$ (**bottom**) depending on maximum speed brake area $A$ for thirteen different aircraft types assuming a clean flap configuration, a Mach number of 0.82, an angle of attack of $\alpha = 0°$, and an altitude of 30,000 ft. Note the different scales on the y-axes.

## 4. Trajectory Modelling with Speed Brakes

The extension of speed brakes and their effects on the descent phase were then investigated using the modelled trajectories. For this purpose, the flight performance model SOPHIA (Sophisticated Aircraft Performance Model) was used. SOPHIA applies the methodology published in [2], but obtains data for the drag polar and the maximum available thrust as a function of altitude and speed from the open-source flight performance model OpenAp [3]. As a reference, a uniform flight plan was set up for all aircraft types, to be compared in order to carry out an analysis limited to the speed brake systems. A flight from cruise to Munich Airport (specifically, the approach point MAGAT 5000 ft, which serves the approach to runway 08L) was set up as a reference flight. For reasons of energy efficiency, the descent should be conducted as long as possible as a continuous descent with idle thrust. However, in controlled airspace, a maximum speed of 250 kt is permitted below an altitude of 10,000 ft. Therefore, before reaching 10,000 ft, a flight segment is usually flown during which the aircraft reduces its descent speed [57]. Since the on-board computers always fly at a speed that is 10 kt below this speed limit [57], a speed of 240 kt is set as the deceleration target speed. This deceleration phase with a maximum flight path angle ($\gamma = 0.5°$) and idle thrust was modelled with SOPHIA at 10,000 ft altitude. To reduce speed in this phase, speed brakes are extended at different speed brake angles $\delta_i$. Per scenario, all speed brakes per wing are extended at the same angle. Figures 6 and 7 show the impact of speed brake angle $\delta_i$ in this deceleration phase on the distance and time of the descent phase on a B777 aircraft. The path angle was calculated from the equilibrium of forces and restricted to 0.5° in the deceleration phase during descent.

As with the reference B747, the speed brakes can be driven to any angle between 0 and the respective maximum angle (see Table 3) when used as speed brakes. An effect

that cannot be reproduced by SOPHIA is the so-called "blowdown effect", in which the speed brakes cannot be extended to these angles despite being set higher, because the air pressure on the surfaces pushes them downwards [58]. To enable a calculation by SOPHIA nevertheless, it is assumed that the individual speed brakes can be set to the maximum values.

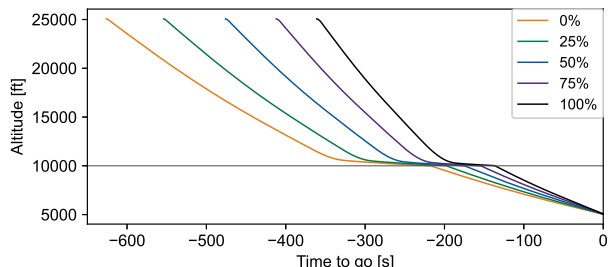

**Figure 6.** Descent profiles of a B777 aircraft as a function of time with different speed brake angles $\delta_i$ (indicated by color). At 25,000 ft, the B777 flies with Mach 0.75; below 10,000 ft 240 kt are mandatory. A maximum deceleration path angle of 0.5° and 9 t payload are considered.

Figure 6 indicates that B777 benefits from its large speed brake area and that it thus achieves a maximum reduction in descent time of 265 s. This results in a reduction of the descent distance of 50.81 km between retracted flaps (orange) and maximum flap position (black). The increased effectiveness of a larger speed brake angle $\delta_i$ induces a displacement of the top of descent towards a later initiation of the descent (see Figure 7).

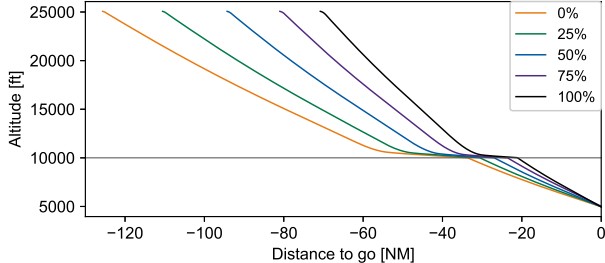

**Figure 7.** Descent profiles of a B777 aircraft as a function of distance at different speed brake angles (indicated by color). A cruising altitude of 25,000 ft, a speed of Mach 0.75 and below 10,000 ft, 240 kt are calculated. A maximum deceleration path angle of 0.5° and a payload of 9 t are considered.

To determine the possible impact of speed brakes on fuel consumption, flights from Dresden starting at a cruising altitude of 25,000 ft are examined in Figure 8. By shortening the descent section, the aircraft remains at the cruising altitude for longer and thus flies with a thrust setting for cruising flight for longer, which is above the idle setting during descent. As a result, the total amount of fuel required for the flight increases. It follows that the use of the speed brakes leads to a lengthening of the cruise segment and this, in turn, leads to a larger amount of fuel required for the entire flight. At the same time, however, a shortening of the descent time becomes possible. The time savings are very small compared to the total flight duration. The reason for this are the steep angles that can be flown by using the speed brakes, especially in the long-haul models. Due to these steep angles, the descent phases have a low horizontal speed component, which is why less distance is covered in these phases. While this also leads to a lengthening of the cruise flight with a greater horizontal speed component, it reduces the savings potential of the total flight duration.

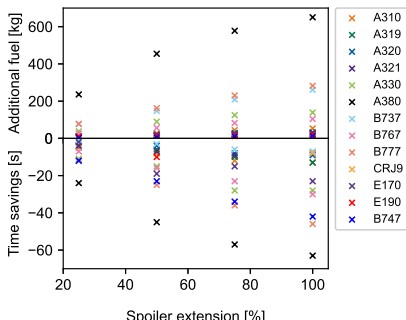

**Figure 8.** Change in fuel consumption (**top**) and time savings (**bottom**) of the entire flight as a function of the speed brake position. (Speed at 25,000 ft: 0.75 Ma; at 10,000 ft: 240 kt; deceleration path angle: 0.5°; payload: 9 t).

The additional fuel consumption for long-haul aircraft (A330 and B777) increases almost linearly with the flap angle used and the resulting larger effective area of the speed brakes (see Figure 9). From this follows an almost linear correlation between change in drag and flap angle for large aircraft. For smaller aircraft models such as the A320 and E170, there is no such linear relationship for large flap angles. Airbus A320 requires additional fuel for the same speed brake area compared to Embraer E170.

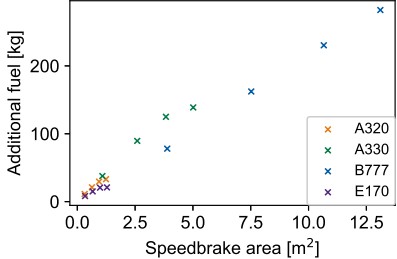

**Figure 9.** Comparison of four Aircraft types regarding the increase in additional fuel consumption depending on effective speed brake area. For initial conditions, see Figure 8.

*Validation*

In addition to flight performance calculations, the model was implemented in an aircraft trajectory optimizer using the robust optimal control theory for continuous descent optimization under real weather conditions [59]. After the new speed brake model was implemented in the flight performance model SOPHIA and in [59], it needed to be validated by modelling a real flight with SOPHIA and comparing the dependent variables, for example, fuel flow, with the real values. Additionally, the aim of validation should have been to prove the benefit in the precision of flight performance modelling, compared to $\Delta c_D = 0.02$ as used in other studies [14]. However, the speed brake angle is not part of usual data sets describing a flight (e.g., ADS-B data). Only Flight Operational Data (FODA) contain this information. FODA remain in the possession of the pilot flying and are actually only released for safety assessments. For this reason, the authors are only in possession of a single data set with extended speed brakes during descent. This flight is by a B748F, which is not significantly different to the B747-100 experimental aircraft, so that the successful transition of the reference aircraft B747-100 to other aircraft types can only be shown to a limited degree. However, since this is the only available data set, it is used here. The authors are happy to validate the model with another data set provided. For this purpose, we modelled a real Boeing B748F flight from Frankfurt, Germany to Boston, USA, operated in March 2018 with speed brakes used in the descent phase. This flight is provided as FODA with wind information (i.e., the difference between ground speed and true airspeed), fuel flow, and speed brake angle, amongst others. Since the study focuses on flight performance with speed brake deflection, we only model the descent phase. This

simplifies the mass estimation of the aircraft, which is not given in FODA. Based on the fuel flow in cruise flight, we determined a payload of 77,705 kg. For the entire flight, we were able to determine a trip fuel of 70,000 kg and thus set the reserve fuel at 10,000 kg (10% contingency plus 3000 kg for holding). For the remaining descent phase, the fuel burn was about 13,500 kg. Thus the total mass of the aircraft is 101,205 kg plus operating empty weight. According to the FODA, speed brakes were deflected at 11,000 ft altitude for 733 s (12.2 min), indicated by speed brake handle up to $\delta = 35°$. Thereby, $v_{TAS}$ was reduced by 79 kt from 248 to 169 kt. Afterwards, the speed brakes were set to $\delta = 4.5°$ for the remaining flight.

As is usual in FODA, thrust is only provided as thrust lever angle [°]. For this reason, for each time step, we calculated the thrust required from the equilibrium of the equations of motion (Equation (4)). Since SOPHIA uses the $v_{TAS}$ as a controlled variable, the aim is to calculate a profile that, despite the lack of lift and drag information in the FODA, reproduces the real FODA flight in terms of temporal and spatial altitude variation as well as fuel consumption. We applied the given speed brake angle $\delta$ equally and synchronously to all speed brakes, whereby the inner speed brake can be extended to a maximum of $\delta = 20°$. Additionally, we modelled the flight in the same way, except for the calculation of the lift and drag coefficients, which were replaced by the common assumption of $\Delta c_D = 0.02$ and $\Delta c_L = 0.00$ [14].This parameter setting is highlighted in orange in Figures 10–12.

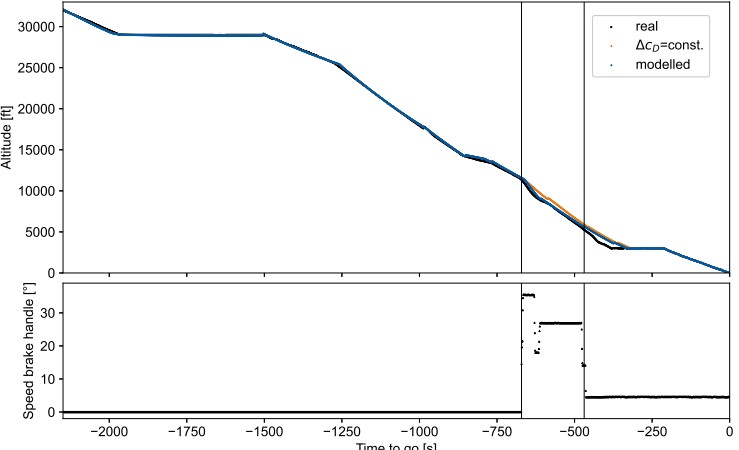

**Figure 10.** Validation of the developed speed brake model: Comparison of altitude over time of a real B748F flight (black) with modelled flights. Blue: using the aircraft-type-specific speed brake model; orange: assuming $\Delta c_D = 0.02$ and $\Delta c_L = 0.00$. Bottom: the speed brake handle provided by the real flight as FODA data. Vertical lines indicate a time frame (12 min) with significantly deflected speed brakes.

With the aircraft-type-specific speed brake model, depending on the angle of attack and flap position, the aircraft model was able to correctly react on speed brake deflection, and follows the real flight profile more closely than the simplified model usually used in flight performance modelling [14] (see Figures 10 and 11), although the profile is not perfectly matched. Specifically, the spontaneous switch from $\delta = 27°$ to $\delta = 4.5°$ (indicated by the second vertical line in Figures 10 and 12) induces deviations from the real flight profile, perhaps due to an overestimation of the controller's inertia in reaching the new target speed, or due to an underestimation of resistance at low speed brake angles. In both modelled profiles (orange and blue), the speed at this point is higher than in real flight, and converges to the real speed at second 1836.

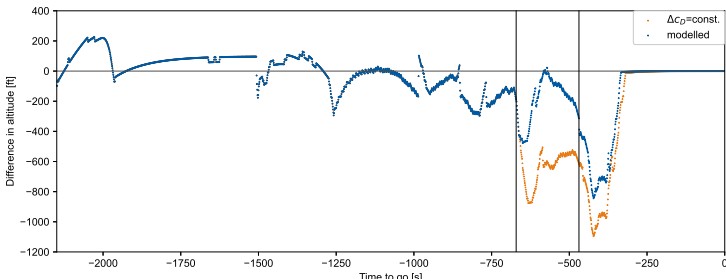

**Figure 11.** Differences in altitude between a real B748F flight and two modelled flights as a function of time. Blue: modelled with the aircraft-type-specific speed brake model; orange: modelled with $\Delta c_D = 0.02$ and $\Delta c_L = 0.00$. Vertical lines indicate a time frame of 12 min with significantly deflected speed brakes.

Besides the vertical difference between the real data and the modelled flights with differences up to 1150 ft in the case of $\Delta c_D = 0.02$ and $\Delta c_L = 0.00$ (orange), Figure 11 shows the ability of SOPHIA's controller to reach a target altitude which is defined every second. Furthermore, the improvement of using the aircraft-type-specific speed brake model (blue) can by quantified to a maximum difference in altitude of 800 ft (Figure 11).

Figure 12 clearly indicates that our model (blue) calculates larger values of $c_D$ as soon as speed brakes are deflected, compared to [14] (orange). Furthermore, the modelled impact on the lift coefficient results in slightly lower values of $c_L$ (mean $c_D = 0.476$) compared to neglecting the speed brake influence on the lift coefficient (orange, mean $c_D = 0.479$). This impact is surprisingly small, maybe due to the short time with strongly deflected speed brakes.

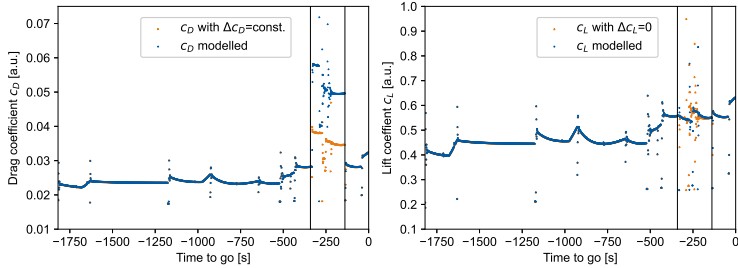

**Figure 12.** Comparison of drag coefficient (**left**) and lift coefficient (**right**) during the B748F validation flight. Blue: modelled with the aircraft-type-specific speed brake model; orange: assuming $\Delta c_D = 0.02$ and $\Delta c_L = 0.00$. The aircraft-type-specific speed brake model results in increased drag and a slightly decreased lift when speed brakes are deflected. Vertical lines indicate time frame with significantly deflected speedbrakes.

## 5. Conclusions

In this study, a rule for the calculation of the effect of speed brakes on flight performance was made possible for twelve aircraft types and applied in trajectory modelling with the flight performance model SOPHIA. For this purpose, a model developed in 1971 for the B747-100 within the framework of a NASA research project was transferred to the other aircraft types. Graphical parameterizations of this speed brake model of the B747 were digitized. With known dimensions of the speed brakes of any aircraft and known profile depth of the wing at the position of the speed brakes, as well as taking into account the possible speed brake angles, corresponding speed brakes and angles on the NASA model B747 could be determined, and these reflect the different operational and performance areas of the respective aircraft models well.

Using Computer-Aided Design (CAD) drawings from aircraft manufacturers, the geometric information of the speed brakes of twelve different aircraft types could be determined with a high degree of reliability. For the transfer of the NASA model to the other

aircraft types, however, restrictions had to first be made to achieve minimal environmental influences on the performance data. Since these limitations are typical for Continuous Descent Operations (CDO) procedures, the accuracy of the results was not significantly affected. However, a closer look at the digitized parameters of the NASA B747 model shows a loss of accuracy, which should be reflected in the determined drag and lift coefficients.

In validation, we modelled a real descent profile with known speed brake deflection during the descent phase and compared the resultant profile following our modelled impact on $c_L$ and $c_D$ with the modelled profile assuming a constant factor of $c_D = 0.02$ for maximum speed brake deflection and $c_L = 0$. Due to unknown information of aircraft mass, inertia, and thrust, we could not accurately follow the real profile, but gained improvements compared to the usual model approach. However, missing real flight performance data including speed brake handle information of other aircraft types hampers our ability to completely validate the model.

The implementation of the effect of speed brakes on lift and drag in the flight performance model SOPHIA now allows a dynamic trajectory optimization, especially in the descent phase. Dynamic changes of both external boundary conditions, such as the influence of the weather or the required arrival time, and internal parameters, such as the cost index, can now be taken into account in the trajectory optimization, even if such changes induce unexpected level flight segments or acceleration phases. We have thus extended the selection of objective functions in SOPHIA and can now devote ourselves to new research questions.

**Author Contributions:** Conceptualization, J.R.; methodology, J.R. and T.S.; software, J.R.; validation, J.R. and T.S.; formal analysis, J.R. and T.S.; investigation, J.R. and G.C.; resources, H.F.; data curation, J.R. and T.S.; writing—original draft preparation, J.R.; writing—review and editing, J.R. and S.K.; visualization, J.R.; supervision, J.R. and H.F.; project administration, J.R.; funding acquisition, J.R. and H.F. All authors have read and agreed to the published version of the manuscript.

**Funding:** This work is a part of the project "Optimized CDO under Uncertain Environmental and Mission Conditions" (project number: 327114631) financed by German Research Foundation (DFG).

**Conflicts of Interest:** The authors declare no conflict of interest.

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
