# Peer review of "Aircraft-Type-Specific Impact of Speed Brakes on Lift and Drag"

_aerospace, doi:10.3390/aerospace9050263_

Round 1

Reviewer 1 Report

In my opinion the validation of the proposal is not finished. The authors recognise that they do not have enough actual flight data to compare with the model results. 

The results of the comparison made are presented in a small graphic in which is difficult to know which is the error/difference between predicted/actual trajectories. 

Author Response

Dear Reviewer 1,

thank you for your excellent ideas to improve the paper.

Please find attached my point-to-point response to all your comments.

With best wishes

Judith Rosenow

Reviewer 2 Report

This manuscript titled “Aircraft-type-specific Impact of Speed Brakes on Lift and Drag” addresses an essential aspect of the trajectory modeling topic. It presented a rule for calculating the effect of speed brakes on flight performance and applied it in the trajectory modeling with the flight performance model SOPHIA. Grammar and spelling are correct. The style is clear and linear, making the manuscript easily accessible and readable.

Aircraft trajectory modeling has been a subject of research for many authors in the last decades. However, the contribution of this paper is not clear and is not adequately identified.

Below I include some further comments or questions if these could be helpful to the authors to improve their paper.

  1. Could it be possible to change the form of the x-axis in Figure 6 and Figure 7? Readers may be more familiar with the “time-to-go” or “distance-to-go” format, especially the altitude profiles in CDO. For example, please refer to Figure 3 in Ref [4] “Robust CDO Trajectory Planning under Uncertainties in Weather Prediction” of the manuscript.
  2. Could it be possible to provide and analyze the range of speed brake angles? In addition, the descent operation should consider passenger comfort, which is dependent on the descent gradient or rate of descent (ROD).
  3.  The lack of field data is the weak point of this manuscript, which results in the validation not being very convincible. The only example is not statistically significant to support any conclusion. Such a concern is why the reviewer has not recommended accepting this paper immediately for publication.
  4.  The other issue is that the contribution of this paper is not clear and is not adequately identified. In other words, the authors did not put forward the potential application of the proposed method. For example, if it applies to the 4D-TP, it is difficult to predict when to extend the speed brake. On the other hand, the proposed method could be used in the trajectory optimization field from the reviewer’s point. Therefore, some additional remarks may be needed in the Conclusion part.

Author Response

Dear Reviewer No. 2,

thank you for your comments. Please find attached our point- to point response.

Round 2

Reviewer 1 Report

Thanks for considering my comments.

My main concern is still about the validation of the proposed method. I understand that there is a lack of actual data. However in my opinion it would be welcome a improved validation exercise.

Reviewer 2 Report

I am satisfied with the revisions made by the authors and I recommend the article to be accepted for publication.